# Estimates of the COVID-19 Infection Fatality Rate for 48 African Countries: A Model-Based Analysis

**Amobi Andrew Onovo [1,2,*], Abiye Kalaiwo [1,2], Christopher Obanubi [3], Gertrude Odezugo [4], Janne Estill [5,6] and Olivia Keiser [6]**

1   Office of HIV/AIDS and TB, U.S. Agency for International Development, Abuja 900211, Nigeria; akalaiwo@usaid.gov
2   Institute of Global Health, Faculty of Medicine, University of Geneva, CH-1211 Geneva, Switzerland
3   Centers for Disease Control and Prevention, Abuja 900211, Nigeria; qvx9@cdc.gov
4   Office of Health, Population and Nutrition, U.S. Agency for International Development, Abuja 900211, Nigeria; godezugo@usaid.gov
5   Institute of Mathematical Statistics and Actuarial Science, Department of Mathematics and Statistics, University of Bern, CH-3012 Bern, Switzerland; Janne.Estill@unige.ch
6   Unit Infectious Disease and Modelling, Faculty of Medicine, Institute of Global Health, University of Geneva, CH-1211 Geneva, Switzerland; Olivia.Keiser@unige.ch
*   Correspondence: amobiandrewonovo@gmail.com; Tel.: +234-7030538954

**Abstract:** (1) Background: Examine global data from 48 African countries to estimate the SARS-CoV-2 infection fatality rate; (2) Methods: We analyzed time series data on the 135,126 confirmed cases and 3922 deaths from COVID-19 disease outbreak in Africa through 30 May 2020. In a Bayesian prediction model based on the Monte Carlo approach, we adjusted for demographic, economic, biological, and societal variables to account for the untested people; (3) Results: We calculated a total of 1,686,879 COVID-19 infections after correcting for possible risk variables in the Bayesian model, equal to 13 infections per confirmed case. In Africa, the IFR is projected to be 0.23% (95% CI: 0.14–0.33%). The percentages varied by country, ranging from 0.004% in Botswana and the Central African Republic to 1.53% in Nigeria. The projected IFR is twelvefold greater than the WHO's 2009 H1N1 influenza pandemic estimate (0.02%). In four countries: Morocco, Nigeria, Cameroon, and South Africa, the inverse distance weighted interpolation map shows high IFR variability; (4) Conclusions: COVID-19 infection mortality rates can vary significantly between regions, and this might be due to changes in demography, underlying health conditions in the community, healthcare system capacity, positive health seeking behavior, and other variables.

**Keywords:** COVID-19; infection fatality rate; Bayesian prediction; Monte Carlo method; influenza; Africa





## 1. Introduction

As of 18 July 2020, 13,876,441 confirmed cases and 593,087 deaths due to coronavirus disease 2019 (COVID-19), caused by the novel severe acute respiratory syndrome coronavirus 2 (SARS-CoV-2), had been reported worldwide [1]. Most persons infected with the COVID-19 virus will have mild to moderate respiratory symptoms and will recover without needing any therapy. People over the age of 65, as well as those with underlying medical conditions such as cardiovascular disease, diabetes, chronic respiratory disease, and cancer, are at a higher risk of developing serious illness [2]. According to global data on COVID-19 disease compiled on 30 May 2020 by the European Center for Disease Prevention and Control (ECDC), the spread had reached 54 countries in Africa with a total of 135,126 laboratory-confirmed COVID-19 cases and 3922 COVID-19 related deaths. Six of Africa's 54 countries (Eritrea, Lesotho, Namibia, Rwanda, Seychelles, and Uganda) had not yet reported any COVID-19 deaths as of 30 May 2020.

To understand the severity of infection during an outbreak, that is, the virulence of the causative agent, the common epidemiological practice is to estimate the case fatality

ratio (CFR) as the risk of death among cases. However, crude CFR was obtained simply by dividing the number of deaths by the number of reported laboratory-confirmed cases, such as those compiled daily by the World Health Organization during the SARS epidemic [3] and those presented on the COVID-19 map dashboard by John Hopkins University, can be misleading [4,5]. During an outbreak of a pandemic or emerging infectious disease such as SARS-CoV-2, the infection fatality rate (IFR) is a more reliable metric to estimate the fatality rate in all the affected countries. The IFR is key to determining the effect of the pandemic at the population level, as well as the effects of public policies and regulations, such as social distancing measures and the effects of potential future shortfalls in health care services.

Knowledge of the IFR of SARS-CoV-2 is necessary to tackle the COVID-19 pandemic [6,7]. The IFR is the ratio of two numbers—the number of deaths caused by COVID-19 (numerator) and the cumulative number of people in the population who were genuinely infected by the virus (denominator). However, for many reasons, both the numerator and the denominator of the IFR are measured with error. For example, errors in the denominator arise because patients remain asymptomatic during the first few days of the infection, testing is not universal and selective at best, and longitudinal data on COVID-19 patients are unavailable at the national level [8]. Because we do not know the true number of people affected, the IFR can be skewed upwards. Because some people who are currently sick may die in the future, or because fatalities are underreported, it can be skewed downward (errors in numerator). During the early stages of testing, the upward bias is likely to be substantially larger. The large disparity in CFR reported by a nation is due largely to testing availability [9,10]. Only hospitalized patients with advanced COVID-19 symptoms were tested in the early stages of the outbreak and in countries where testing was limited. Because many illnesses in the community are undiagnosed, the CFR is an exaggerated estimate of the IFR [11,12].

There are relevant studies on seroprevalence that are useful for estimating the number of infections at the community level, as well as provide robust IFR estimates when these data are triangulated with the number of deaths. Recent serosurvey for the canton of Geneva, Switzerland using a Bayesian regression model estimated a population-wide IFR of 0.64% (0.38–0.98) [13]. Gudbjartsson et al. recently published research estimating the prevalence of COVID-19 fatalities in Iceland using Bayesian analysis [14]. For Spain as a whole, the infection fatality rate was 1.15% and ranged from 0.13–3.25% across 19 Spanish regions [15]. Another study examined the seroprevalence of antibodies to SARS-CoV-2 in a community sample drawn from Santa Clara County, California. The study reported an IFR of 0.17% [16]. Such findings show that the IFR for SARS-CoV-2 varies across countries and regions. At the time of this analysis, there were no reports of seroprevalence studies conducted in Africa. In this paper, to effectively estimate the IFR of COVID-19 for 48 African countries, we provide a new statistical approach for eliminating measurement errors in the denominator. We attempted to account for people with undetected COVID-19 disease in the denominator, that is, untested individuals by adjusting for the underlying socio-demographic, economic, and potential biological risk factors in a Bayesian statistical model using laboratory-confirmed reported cases of COVID-19 as the response variable in our model. Because of its potential to use prior information or experimental evidence (e.g., risk factors correlated with COVID-19) in a data model, we used Bayesian statistical modeling to produce more realistic outcomes (i.e., estimated number of people infected with COVID-19). Consequently, we calculated the IFR by dividing the total number of reported deaths by the adjusted or estimated denominator.

## 2. Materials and Methods

### 2.1. Setting and Data Sources

Africa accounts for about 16% of the world's human population with 1.3 billion people as of 2018 [17]. We gathered and analyzed data on the 135,126 confirmed cases and 3922 deaths from the COVID-19 disease outbreak across Africa between 15 February 2020 through 30 May 2020. We used publicly documented COVID-19 datasets created by Our

World in Data (https://covid.ourworldindata.org/data/owid-covid-data.csv, accessed on 28 April 2021) and utilized time-series aggregate data compiled by ECDC on the total number of confirmed cases, total number of deaths, and other variables of potential interest. The following other variables were collected: population density (number of people divided by land area, measured in square kilometers), the proportion of people aged 65 and above, access to handwashing facilities (proportion of the population with basic handwashing facilities on the premises), socio-economic situation (proportion of people living in extreme poverty), diabetes prevalence (among the population aged 20 to 79 years), death rate from cardiovascular disease, and the transmission classification type for COVID-19 infection categorized into community transmissions, clusters of cases and sporadic cases. We excluded six of the 54 African countries from this study, which at the time of this analysis had not reported any COVID-19 deaths yet.

## 2.2. Assumptions

Our analysis is backed by the following assumptions: (1) That numerator and denominator errors can lead to an under-reporting of actual SARS-CoV-2 deaths and infections, though the error margin for deaths is smaller than for infections; (2) The estimated cumulative number of SARS-CoV-2 infections represents the IFR calculation denominator. In our study, the estimated cumulative number of COVID-19 infections was summarized using the Bayesian posterior summary statistics; mean, median, 95% lower credible interval, and 95% upper credible interval. To obtain deeper insights into the uncertainty around our estimates, we examined the cumulative estimated COVID-19 infections over a range of mean and maximum posterior summary statistics (75%, 90%, and 95%) through sensitivity analysis (Appendix A.1). The calculated IFR using each summary statistic was compared with the IFR from recent seroprevalence surveys (see Appendix A.1: sensitivity analysis); (3) We assume that the severity of COVID-19 depends on the covariates in our model (Figure 1). Assumption #1: during the early stages of an epidemic, both mortality and actual infections are undercounted [18,19]. We believe that, at any given moment, the mistakes in the denominator are higher than the inaccuracies in the numerator since fatalities are far more apparent occurrences than illnesses; Assumption #2: our central assumption that adjusts for biases in the denominator. Here we posit that the denominator, that is, the estimated cumulative number of individuals infected with COVID-19, represents the estimated upper limit of the 95% credible interval that we use as a proxy for the actual cumulative number of infections. Assumption #3: our constructed conceptual framework that acts as a bridge between model adjustments for the denominator and empirical observations.

## 2.3. Statistical Model

We used the Bayesian parametric model to predict the cumulative number of individuals infected with COVID-19 as of 30 May 2020. Firstly, we fitted a Bayesian normal regression model using Gibbs sampling based on the technique of the Markov Chain Monte Carlo (MCMC) to specify the posterior model. The parameters were then estimated iteratively until the burn-in conditions were met.

The Bayes' rule can be written as:

$$\text{Posterior} \propto \text{Likelihood} \times \text{Prior} \qquad (1)$$

The equation for Bayesian normal regression with the response sampled from a normal distribution is:

$$y \sim N\left(\beta^T X, \sigma^2 I\right) \qquad (2)$$

The likelihood for the model is defined as the joint probability of observing the data given the parameters and is given by:

$$p(\text{y} \mid \boldsymbol{\beta}, \sigma^2) = \text{N}(\boldsymbol{X\beta}, \sigma^2 \text{I}) = (1/2\pi\sigma^2)^{\text{n}/2} \exp\{-1/2\sigma^2 (\text{y} - \boldsymbol{X\beta})^T (\text{y} - \boldsymbol{X\beta}) \tag{3}$$

The inverse gamma distribution (or Gaussian-inverse-gamma distribution):

$$p(\boldsymbol{\beta}, \sigma^2) = p(\sigma^2)\, p(\boldsymbol{\beta} \mid \sigma^2) \tag{4}$$

where $p(\sigma^2)$ is an inverse-gamma distribution

$$p(\sigma^2) \propto (\sigma^2) - \text{v}^0/2 - 1 \exp(-\text{v}_0\, \text{s}^0/2\, \sigma^2) \tag{5}$$

The posterior probability distribution is written as:

$$p(\boldsymbol{\beta}, \sigma^2 \mid \boldsymbol{Y}, \boldsymbol{X}) \propto p(\boldsymbol{Y} \mid \boldsymbol{X}, \boldsymbol{\beta}, \sigma^2)\, p(\sigma^2)\, p(\boldsymbol{\beta} \mid \sigma^2) \tag{6}$$

$$p(\boldsymbol{\beta}, \sigma^2 \mid \boldsymbol{Y}, \boldsymbol{X}) \propto (\sigma^2)^{-n/2} \exp[-1/2\sigma^2(\boldsymbol{Y} - \boldsymbol{X\beta})^T (\boldsymbol{Y} - \boldsymbol{X\beta})] \times (\sigma^2) - (v/2+1) \exp[-vs^2/2\sigma^2] \times (\sigma^2)^{-k/2} \exp[-1/2\sigma^2(\boldsymbol{\beta} - \boldsymbol{\mu})^T \Lambda(\boldsymbol{\beta} - \boldsymbol{\mu})] \tag{7}$$

where:

$\boldsymbol{Y}$ = dependent variable
$\boldsymbol{X}$ = the matrix of independent variable
$\boldsymbol{\beta}$ = vector of regression model parameters
$\sigma^2$ = Standard deviation
$\boldsymbol{\mu}$ = prior mean $\boldsymbol{\mu}$
$\Lambda$ = prior precision matrix
$k$ = number of regression coefficients
V = prior hyperparameter values

The posterior model combines a probability distribution, which contains information about model parameters based on seen data, with a prior function that contains previous information about model parameters (before viewing the data). The model parameters included the response variable "confirmed reported cases of COVID-19 across Africa" and the independent covariates of interest; population density, age 65 or above, cardiovascular death rate (CVD), diabetes prevalence, handwashing facilities, and extreme poverty. Secondly, we computed Bayesian predictions for the outcome variable. Based on results from the fitted posterior model, we estimated the cumulative number of individuals infected with COVID-19 using the "bayespredict" model in StataMP v.16. Here we simulated 1000 MCMC samples of outcome values for each of the 48 countries and calculated the posterior means and estimated p-values for each observation. We used a random-number seed to ensure reproducibility. Finally, we conducted posterior estimated checks by comparing the observed data with the MCMC replicates (simulated data from the posterior predictive distribution). Unlike classical prediction, which produces a single value for each observation, Bayesian prediction produces an MCMC sample of values for each observation.

### 2.4. Spatial Analysis

We georeferenced the estimated IFR across 48 countries and performed two analyses. We used the inverse distance weighted interpolation (IDW) technique in the Geostatistical Analyst tool of ArcGIS 10.8 software to create a raster showing the spatial distribution of COVID-19 IFR. Second, we constructed a thematic map contrasting the estimated COVID-19 IFR and Influenza IFR 2018–2019 to obtain clear insights into the severity of the COVID-19 outbreak.

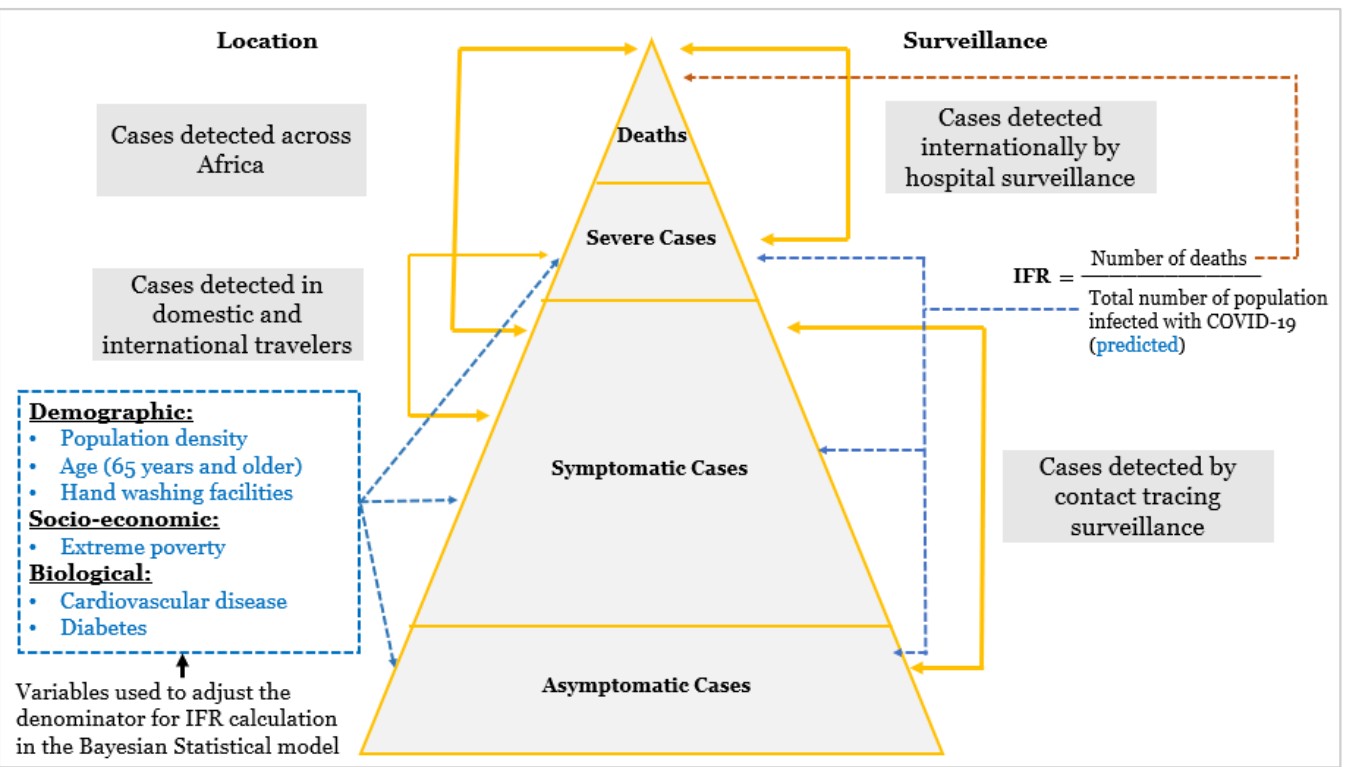

**Figure 1.** Categorization of SARS-CoV-2 cases according to severity and variables included in the Bayesian model.

## 3. Results

### 3.1. Descriptive Analysis

Our analysis was based on 48 of the 54 African countries that reported confirmed cases and deaths from coronavirus between 15 February 2020 through 30 May 2020. At the time of this analysis, nine countries (Egypt, Algeria, South Africa, Nigeria, Sudan, Morocco, Cameroon, Mali, and Somalia) accounted for 80% of Africa's COVID-19 deaths (Figure 2a). The scatter plot shows an uphill pattern from left to right; this indicates a strong positive correlation between total deaths due to COVID-19 and total confirmed cases of COVID-19, ($r_s$ (47) = 0.92, $p$ = 0.001) (Figure 2b). The scatter plot illustrates the variability in the pattern of reported deaths and confirmed cases throughout the nine countries. The result is suggestive that the risk of death among cases varies by location and is typically changing over time. Among these countries, the calculated crude CFR was the highest in Algeria (7.0%) followed by Mali (6.0%) and the lowest in South Africa (2.1%). The lower crude CFR for COVID-19 in South Africa compared with the other eight countries may be caused by differences in demographics, socio-economic and biological characteristics. This could also suggest variability in the testing capacity of COVID-19 across these countries.

#### 3.1.1. Bayesian Regression Model

The summary of the fitted Bayesian multiple linear regression model is provided in Table 1. The response variable was log-transformed to control for skewness and ensure an effective linear relationship with the explanatory variables. The default priors used for the model parameters were; normal (mean 0, standard deviation 10,000) for the regression co-efficients and inverse gamma (shape 0.01, scale 0.01) for the variance parameter. In Table 1, the first two columns of the Bayesian normal regression report the posterior means and standard deviations of the model parameters. The posterior means and standard deviations of the regression coefficients were remarkably similar to the least-square estimates. The posterior mean estimate for the variance, 2.20, was close to the residual mean squared estimate, 2.15. The minimum efficiency in the model was 0.76, and the mean efficiency 0.97. Our acceptance rate (AR) was good, and efficiencies were high. We did not have a reason to sus-

pect nonconvergence. Nevertheless, we explored convergence by computing graphical diagnostic plots for all models to confirm this. Overall, graphical diagnostic plots showed that MCMC converged and mixes well for all parameters in the model (see graphical diagnostic plots in Appendix A.2). The STATA website has a link to the Bayesian regression model used in this study at https://www.stata.com/features/overview/bayesian-predictions/ (accessed on 28 April 2021).

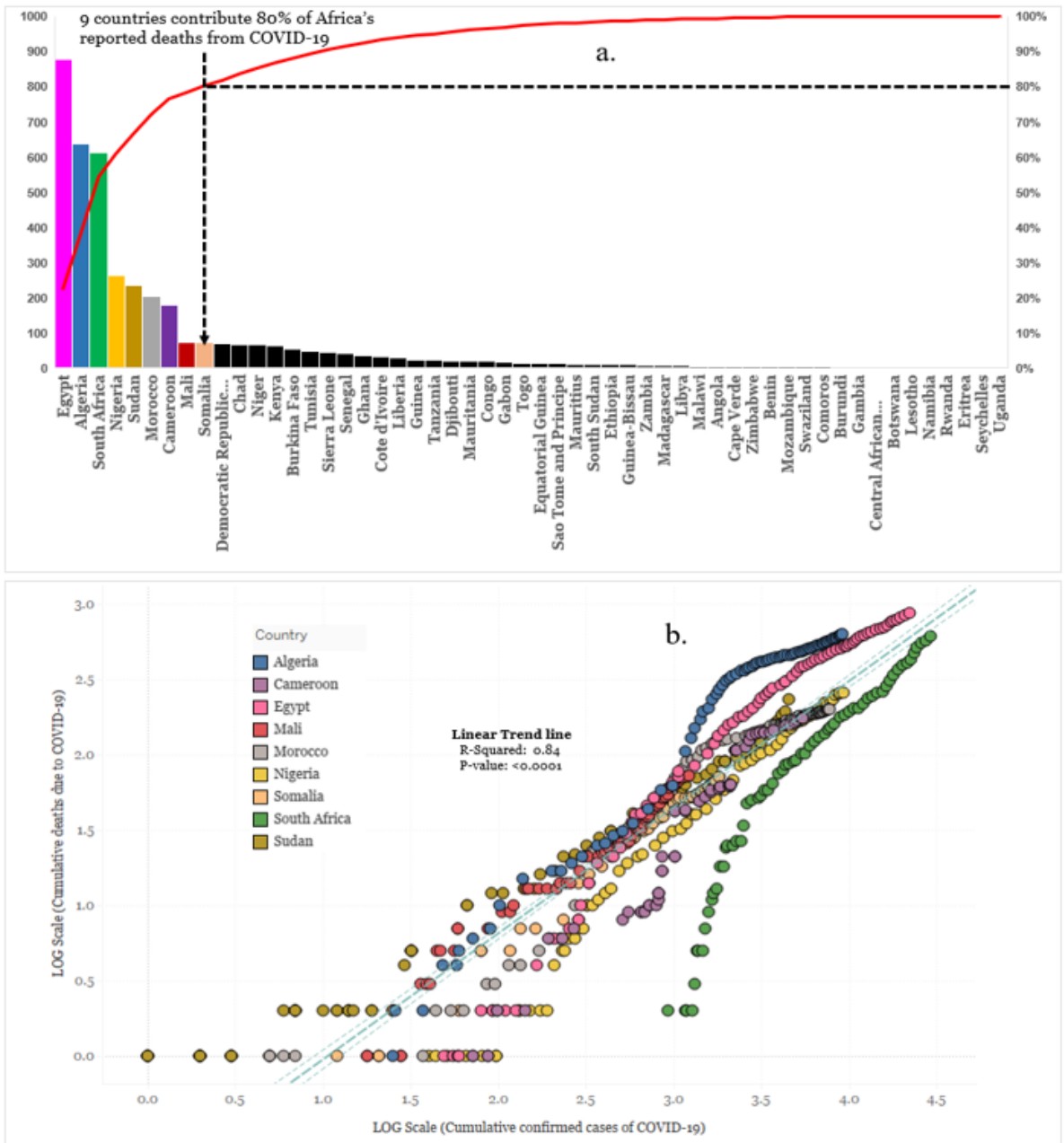

**Figure 2.** (**a**) Reported COVID-19 deaths in each African country (**b**) Cumulative confirmed COVID-19 deaths vs. cases for the nine countries that represent 80% of reported deaths, 30 May 2020. The several points in (**b**) represent the multiple times of analysis.

### 3.1.2. Bayesian Prediction Model

We calculated the posterior summary statistics for all simulated outcome observations (Appendix A.3) and computed the posterior predictive summaries to test our prediction. The simulated outcome values were saved in a prediction dataset. The mean, minimum, and maximum test statistics agreement between the expected and observed data was

compared using the prediction results (Appendix A.4). The mean statistic had a posterior predictive p-value of 0.46, the lowest was 0.38, and the highest was 0.69. When this probability is close to 0.5, the replicated and observed data agree with respect to the test statistic. However, Gelman et al. [20] indicated that values between 0.05 and 0.95 are often considered acceptable. According to our results, the mean, minimum, and maximum statistics appear to agree between the observed and replicated data.

**Table 1.** Bayesian normal regression using Gibbs sampling. MCMC iterations = 3500; Burn-in = 2500; MCMC sample size = 1000; Number of obs = 48; Acceptance rate = 1; Efficiency: min = 0.7658; avg = 0.9707; max = 1.

|  | Mean | Std.Dev | MCSE | Median | Equal-Tailed 95% Cred. Interval | |
| --- | --- | --- | --- | --- | --- | --- |
| Confirmed_Cases | | | | | | |
| Population Density | −0.0052613 | 0.0021246 | 0.000067 | −0.0051941 | −0.009677 | −0.0011185 |
| Aged 65 Older | 0.1831922 | 0.1896953 | 0.005703 | 0.1855481 | −0.174675 | 0.550733 |
| Cvd Death Rate | −0.0011224 | 0.0036282 | 0.000113 | −0.0011888 | −0.008117 | 0.0061924 |
| Diabetes Prevalence | 0.062837 | 0.0799469 | 0.002528 | 0.0618234 | −0.093096 | 0.2195652 |
| Handwashing Facilities | 0.0200225 | 0.0102218 | 0.000319 | 0.0200439 | 0.0006078 | 0.0405081 |
| Extreme Poverty | −0.0028848 | 0.010585 | 0.000335 | −0.0029234 | −0.024479 | 0.017851 |
| _Cons | 6.22023 | 1.139261 | 0.036027 | 6.19992 | 3.77586 | 8.390462 |
| Var | 2.209552 | 0.5010807 | 0.018107 | 2.147209 | 1.413217 | 3.442549 |

MCMC iterations are random samples from the posterior means, and Burn-in is the number of iterations thrown away at the beginning of the MCMC run. The number of MCMC draws used to calculate the Bayesian credible bounds is known as the MCMC sample size. The acceptance rate refers to the percentage of simulations that were adopted, while efficiency refers to how well the model performed.

### 3.1.3. Estimated Infection Fatality Rate by Country

We estimated 1,686,879 cumulative SARS-CoV-2 infections, which is the denominator used in our study to calculate the IFR. In order to measure the IFR, we divided the total number of deaths reported (3922) by the estimated denominator. The estimated overall IFR in the 48 African countries was 0.23% (Std. Dev: 0.04%, 95% confidence interval 0.14–0.33%) as of 30 May 2020. The confidence intervals around the IFR were calculated using the normal-based confidence intervals in StataMP v.16. The rates varied from 0.004% in Botswana and the Central African Republic to 1.53% in Nigeria, respectively. Fourteen African countries: Nigeria, South Africa, Cameroon, Morocco, Niger, Burkina Faso, Sierra Leone, Democratic Republic of Congo, Egypt, Kenya, Somalia, Chad, Sudan, and Senegal represent 80% of the total cumulative deaths among the infected population, with Nigeria presenting the highest IFR (Table 2). In terms of the COVID-19 transmission scenario, a majority at 27 (56%) of the African countries reported community transmission as the major way coronavirus is spreading. Furthermore, 18 (38%) countries indicated that most cases were observed in clusters, while Ghana, South Africa and Zimbabwe indicated only sporadic COVID-19 cases without notable community transmission or clusters (Table 2).

An interpolated map displaying high, moderate, and low rates of COVID-19 IFR spread across Africa, using the IDW technique, is shown in Figure 3 (Left panel). The map shows the variability of high IFR in four countries: Morocco, Nigeria, Cameroon, and South Africa. IFR is moderately high in specific regions in Northern and Western Africa, whereas predominantly low in the southern, central, and eastern regions, except for countries such as the Democratic Republic of Congo, Kenya, and Somalia (Horn of Africa) with comparable rates to regions with high IFR. Figure 3 (Right panel) compares estimated COVID-19 IFR and Influenza IFR 2018–2019 (0.1%) [21] by country. Countries shaded black have COVID-19 IFR above 0.1% (which is the 2018–2019 Influenza IFR) and countries shaded blue have COVID-19 IFR below 0.1%.

**Table 2.** Total Estimated Infection Fatality Rate for 48 African Countries as of 30 May 2020.

| Country | Total Cases Reported | Total Deaths Reported | Cumulative Infections Estimated | Estimated IFR | Crude CFR | COVID-19 Transmission Classification Type |
|---|---|---|---|---|---|---|
| Algeria | 9134 | 638 | 272,017 | 0.24% | 7.00% | Community transmission |
| Angola | 77 | 4 | 19,187 | 0.02% | 5.20% | Clusters of cases |
| Benin | 224 | 3 | 8884 | 0.03% | 1.30% | Community transmission |
| Botswana | 35 | 1 | 22,819 | 0.00% | 2.90% | Clusters of cases |
| Burkina Faso | 847 | 53 | 10,257 | 0.52% | 6.30% | Community transmission |
| Burundi | 42 | 1 | 1959 | 0.05% | 2.40% | Clusters of cases |
| Cameroon | 5436 | 177 | 17,603 | 1.01% | 3.30% | Clusters of cases |
| Cape Verde | 405 | 4 | 12,464 | 0.03% | 1.00% | Community transmission |
| Central African Republic | 874 | 1 | 23,043 | 0.00% | 0.10% | Clusters of cases |
| Chad | 759 | 65 | 17,517 | 0.37% | 8.60% | Community transmission |
| Comoros | 87 | 2 | 4894 | 0.04% | 2.30% | Community transmission |
| Congo | 587 | 19 | 40,170 | 0.05% | 3.20% | Community transmission |
| Cote d'Ivoire | 2750 | 32 | 10,386 | 0.31% | 1.20% | Community transmission |
| Democratic Republic of Congo | 2833 | 69 | 14,499 | 0.48% | 2.40% | Community transmission |
| Djibouti | 2914 | 20 | 23,332 | 0.09% | 0.70% | Clusters of cases |
| Egypt | 22,082 | 879 | 205,083 | 0.43% | 4.00% | Clusters of cases |
| Equatorial Guinea | 1043 | 12 | 39,917 | 0.03% | 1.20% | Community transmission |
| Ethiopia | 968 | 8 | 19,147 | 0.04% | 0.80% | Clusters of cases |
| Gabon | 2613 | 15 | 27,858 | 0.05% | 0.60% | Clusters of cases |
| Gambia | 25 | 1 | 5357 | 0.02% | 4.00% | Community transmission |
| Ghana | 7616 | 34 | 21,248 | 0.16% | 0.40% | Sporadic cases |
| Guinea | 3656 | 22 | 16,782 | 0.13% | 0.60% | Community transmission |
| Guinea-Bissau | 1256 | 8 | 8235 | 0.10% | 0.60% | Community transmission |
| Kenya | 1745 | 62 | 15,440 | 0.40% | 3.60% | Community transmission |
| Liberia | 273 | 27 | 11,455 | 0.24% | 9.90% | Community transmission |
| Libya | 118 | 5 | 25,774 | 0.02% | 4.20% | Community transmission |
| Madagascar | 698 | 5 | 30,097 | 0.02% | 0.70% | Clusters of cases |
| Malawi | 273 | 4 | 5695 | 0.07% | 1.50% | Clusters of cases |
| Mali | 1226 | 73 | 42,636 | 0.17% | 6.00% | Clusters of cases |
| Mauritania | 423 | 20 | 18,496 | 0.11% | 4.70% | Community transmission |
| Mauritius | 335 | 10 | 19,972 | 0.05% | 3.00% | Clusters of cases |
| Morocco | 7714 | 202 | 25,380 | 0.80% | 2.60% | Clusters of cases |
| Mozambique | 234 | 2 | 11,812 | 0.02% | 0.90% | Clusters of cases |
| Niger | 955 | 64 | 12,248 | 0.52% | 6.70% | Clusters of cases |
| Nigeria | 9302 | 261 | 17,052 | 1.53% | 2.80% | Community transmission |
| Sao Tome and Principe | 463 | 12 | 10,292 | 0.12% | 2.60% | Community transmission |
| Senegal | 3429 | 41 | 13,239 | 0.31% | 1.20% | Clusters of cases |
| Sierra Leone | 829 | 45 | 8729 | 0.52% | 5.40% | Community transmission |
| Somalia | 1828 | 72 | 19,366 | 0.37% | 3.90% | Community transmission |
| South Africa | 29,240 | 611 | 47,859 | 1.28% | 2.10% | Sporadic cases |
| South Sudan | 994 | 10 | 26,287 | 0.04% | 1.00% | Community transmission |
| Sudan | 4521 | 233 | 71,606 | 0.33% | 5.20% | Clusters of cases |
| Swaziland | 279 | 2 | 22,122 | 0.01% | 0.70% | Community transmission |
| Tanzania | 509 | 21 | 41,536 | 0.05% | 4.10% | Community transmission |
| Togo | 428 | 13 | 9720 | 0.13% | 3.00% | Community transmission |
| Tunisia | 1071 | 48 | 302,601 | 0.02% | 4.50% | Community transmission |
| Zambia | 1057 | 7 | 14,677 | 0.05% | 0.70% | Community transmission |
| Zimbabwe | 160 | 4 | 20,130 | 0.02% | 2.50% | Sporadic cases |

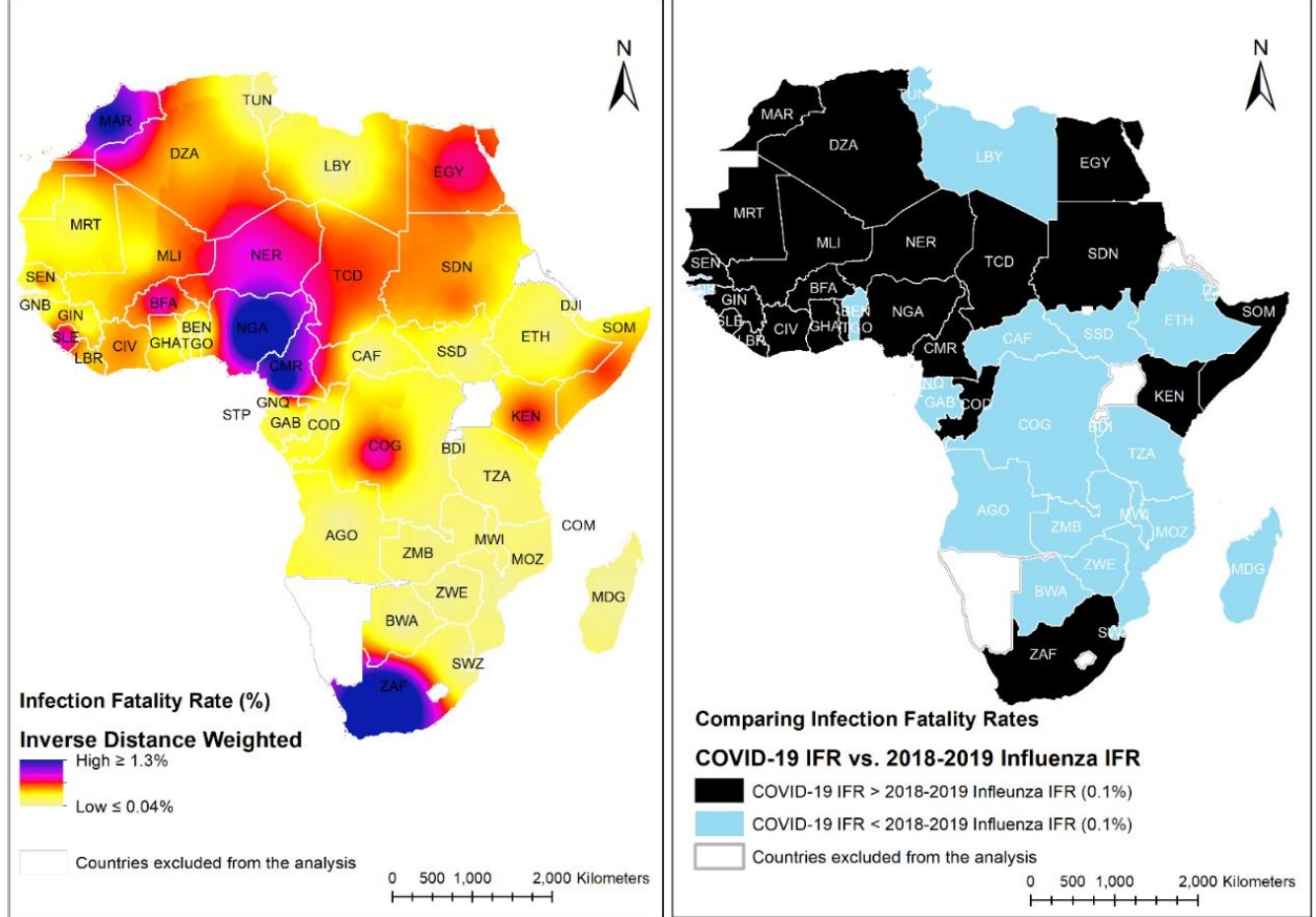

**Figure 3.** (**Left panel**)-Interpolated map displaying high, moderate, and low rates of COVID-19 Infection Fatality Rate spread across Africa, using inverse distance weighted interpolation technique; (**Right panel**)-COVID-19 IFR by country vs. Influenza IFR 2018–2019 (0.1%); Black color represents countries with COVID-19 IFR above 0.1% (which is the 2018–2019 Influenza IFR) and blue color represents countries with COVID-19 IFR below 0.1%.

The log-linear relationship between cumulative number of infections and the explanatory variables in our analysis indicates that the cumulative number of COVID-19 infection increases with age, basic handwashing facilities, diabetes prevalence and cardiovascular disease death rate (Figure 4). Case transmissions classified as clusters and community, according to the WHO COVID-19 categorization (Annex 4) appear to be distributed similarly in terms of population, age, diabetes prevalence, and hand washing facilities.

The log-linear relationship between cumulative number of infections and the explanatory variables suggests a negative moderate and negative weak association between the cumulative number of infections and extreme poverty and population density, respectively. The separate color markers (red = clusters of cases, green = community transmission, and blue = sporadic cases) represent the different countries' COVID-19 transmission classification types. The blue line denotes the regression estimate of IFR as a function of the explanatory variables, and the shaded region depicts the 95% confidence interval for that estimate. Please see Annex 4 for details on the definition of the type of COVID-19 transmission classification type.

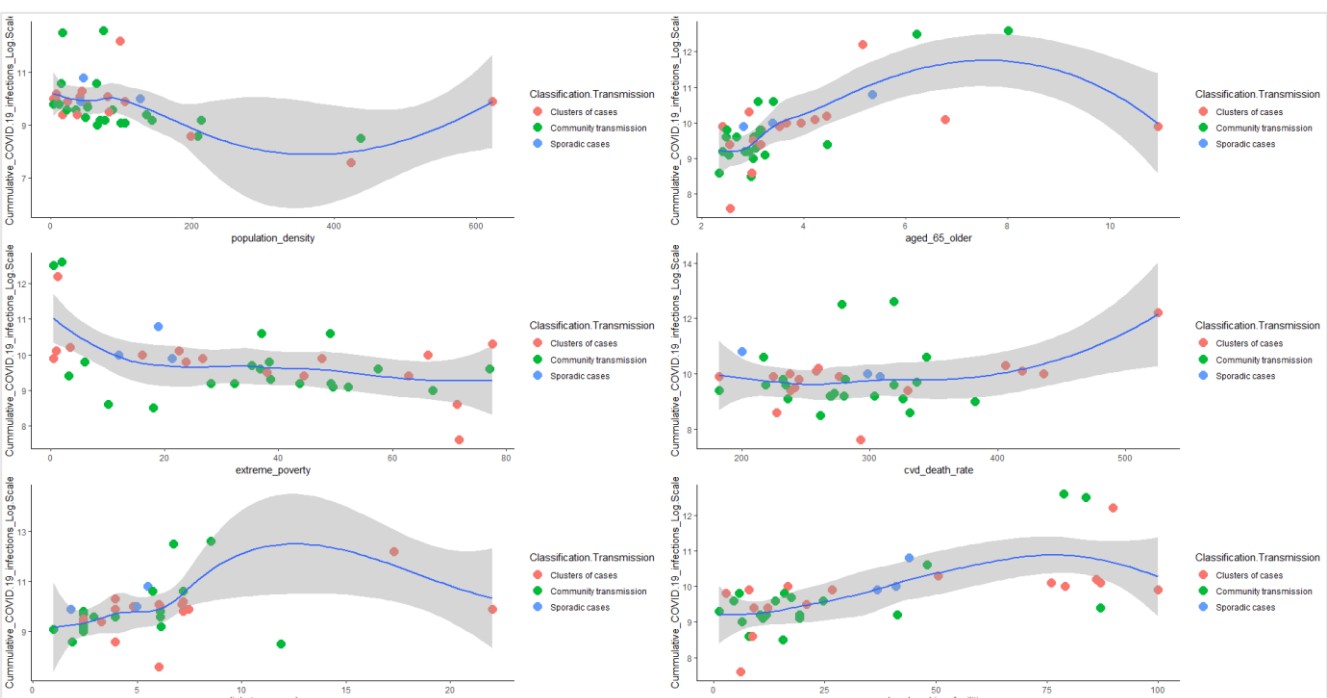

**Figure 4.** The log-linear relationship plots between the cumulative number of infections and the explanatory variables.

## 4. Discussion

From extensive analysis of data from different regions of Africa, our best estimate for the IFR of COVID-19 in 48 African countries at the end of May 2020 is 0.23% (95% CI: 0.14–0.33%). Although this value was lower than China's overall IFR calculation of 0.66% (95%CI: 0.39–1.33%) as reported by PCR testing of foreign residents of Wuhan returning on repatriation flights [22], it is 12-fold higher than the WHO-reported estimate (0.02%) from the 2009 H1N1 influenza pandemic [23,24]. Our findings seem to align with estimates observed in a recent seroprevalence study showing IFR estimates ranging from 0.02–0.40% [18]. In Iceland, the country with the highest number of tests per capita in 2020, the IFR lies somewhere between 0.03% and 0.28% [25]. IFR varied disproportionality across African countries. A total of 15 countries had IFRs higher than the overall estimate of 0.23%, ranging from 0.31–1.53%. For these 15 countries, the mean IFR was comparable with the overall IFR from China (0.61% vs. 0.66%). Two countries, Liberia, and Algeria with IFR 0.24% mirror the continent-wide average, and 31 countries have IFRs ranging from 0.004% to 0.17% lower than the continent-wide average. In 23 (48%) countries, the IFR of SARS-C0V-2 was higher than the IFR of influenza in 2018–2019. A comparison of the two maps in Figure 3 (interpolated map on the left and the thematic map on the right) displays a near glove fit or mirror image. The different IFRs across Africa probably indicate that countries were at different stages of the pandemic, but various other factors may also be important. These include for example underlying health issues in the population, and differences in demographics (e.g., detailed population age structure), in health care systems, in testing practices (including testing practices among diseased persons), and differences in the capacity in responding to the pandemic.

The estimated IFR for each summary statistic was compared with the IFR and the cumulative cases of COVID-19 per confirmed case from recent seroprevalence surveys conducted in Geneva, Spain, and Santa Clara Country, California. The calculated IFR using mean statistics was approximately 8.2%, 6.9%, and 6.2% in the 75%, 90%, and 95% credible intervals, respectively, while the calculated IFR using maximal statistics was 0.31%, 0.26%, and 0.23% in the 75%, 90%, and 95% ranges, respectively. The calculated IFR using the mean statistics was an 11-fold-increase compared to the reported population-wide IFR of 0.64% in Geneva on average, and a six-fold increase compared to the estimated IFR of

1.15% in Spain. This was 11 times higher than the IFR of 0.17% reported in Santa Clara County, California. On average, the calculated IFR using the maximum statistics was moderately lower than the IFR of 0.64% reported in Geneva. While the maximum IFR was five times smaller than the estimated IFR of 1.15% in Spain, the maximum IFR aligned markedly with the 0.17% IFR reported in Santa Clara County, California. The estimated cumulative number of infections of the maximum statistics were 1,265,159, 1,518,191, and 1,686,879 over the three credible intervals, approximately corresponding to 9, 11, and 13 number of infections per confirmed case. In terms of the mean statistics, the estimated cumulative number of infections were 47,366, 56,839, and 63,154, corresponding to less than one infection per confirmed case. Across the three credible intervals of the mean statistics, the number of infections per confirmed case was substantially lower. That is a much smaller share of unreported infections compared to the maximum statistics. The reported overall number of infections in a seroprevalence study performed in Stockholm, Sweden was 74,089, corresponding to 44 infections per confirmed case [26]. We estimated a total of 11 infections per confirmed case in the Geneva serosurvey and 54 infections per confirmed case in the Clara County, California survey. Such findings indicate geographical variation in the distribution of SARS-CoV-2 infections across the globe. This is also an indicator that certain countries are experiencing far more rapid and broader transmission of SARS-CoV-2 infections than others. Since COVID-19 is extremely contagious and a single case will infect dozens of people, we present in this analysis the overall cumulative COVID-19 infections estimated as 1,686,879, corresponding to a total of 13 infections per confirmed case. Our result is slightly higher than the total number of infections per confirmed case presented in the Geneva study, and it resonates with the larger number of infections per confirmed case as seen in the other serosurveys.

Our approach was to control for the upward bias. We did not control for the downward bias that may arise because some of the detected cases may die in the future or COVID-19 deaths may remain unregistered/misclassified. Since the first reported case in Egypt on 14 February 2020 [27], several African countries have intensified efforts to reduce infections and prevent coronavirus deaths by restriction in movements through curfews and lockdowns, deploying trained public health forces, expanding public health surveillance activities to identify all suspected cases, setting up facilities to isolate and treat patients, and ramping up testing capacity. In this study, we attempted to account for a proportion of the untested individuals in the denominator by adjusting for proportions of populations 65 years and older, population density, availability of facilities for basic handwashing, prevalence of diabetes, mortality rate from CVD, and extreme poverty. These variables were included in our analyses because there are so many demographic and socio-economic factors that are attributed as determinants for the different coronavirus-related outcomes. Several studies report that an elderly population [28,29] and an underdeveloped health care system [30,31] are characteristics of countries that take the hardest hit. In an epidemiological study, Safiya et al. found that among 282 patients needing mechanical ventilation, 97.2% of the patients aged 65 or above died [32]. According to the WHO, inadequate housing and overcrowding are major factors in disease transmission, and disease outbreaks become more frequent and extreme when there is a high population density [33]. Studies suggest that elderly adults with clinical comorbid illnesses, such as diabetes and cardiovascular disease, are at higher risk of hospitalization and COVID-19 death [34,35]. To avoid infectious diseases like COVID-19, handwashing with soap and water on a regular basis is necessary. According to recent estimates, 3 billion people throughout the world do not have access to soap or water at home, 900 million children do not have access to soap and water at school, and up to 40% of health care institutions lack hand hygiene equipment [36]. On average, according to our analysis, the proportion of the population in Africa with access to water and soap hand-washing facilities is 35% (95% CI: 26–44).

Since coronavirus spreads by human interaction [37,38], it is a widely accepted assumption that dense populations make for the faster spread of COVID-19, however, an analysis conducted in the USA at Johns Hopkins Bloomberg School of Public Health of

913 urban counties show that infection rates are not related to population density. This resonates with the findings from our Bayesian regression analysis which suggests an inverse relationship between COVID-19 infection and population density. African countries were among the first to identify and report the first cases of coronavirus and enacted and implemented lockdown policies and curfews in March and April 2020 to curb the spread of COVID-19. The inverse COVID-19 infection relationship with population density could be attributed to the stringent public health measures implemented to stymie the outbreak. Prior research on the relationship between poverty and epidemics has found a substantial positive correlation between poverty and the proportion of the population with infectious illnesses in different countries [39,40]. While poverty is frequently thought to be a cause of disease, the nature of the link between poverty and communicable diseases suggests a more complicated relationship [41,42]. A modeling study by Anand et al. used a simple network model to study the fractions of poor and non-poor infected persons during an epidemic under different kinds of interventions. The study found that in the absence of intervention, peak infection caseloads are maximized, but there are no variations in infection levels between poor and non-poor people. Another study by Chie et al. investigated the effect of COVID-19 on poverty and living standards in Ghana [43]. The study went on to look at which groups of people within the income distributions were the worst impacted by the epidemic. The findings reveal that coronavirus has a greater impact on the middle and upper classes in terms of overall household spending than it does on the lower classes. These results provide perspective on the observed inverse relationship between COVID-19 infections and poverty in our study.

In the African region, only one can provide high-quality cause-of-death data (Mauritius), with another three able to provide low or medium-quality data (Seychelles, South Africa, and Zimbabwe). In addition, Egypt and Morocco can provide low to medium-quality cause-of-death data [44]. In our study, we note that there might have been underreporting of coronavirus-related deaths across certain African countries. However, given the early surveillance systems and measures instituted by countries to stymie the outbreak, we believe registration of deaths from COVID-19 would have improved over time. Governments across Africa are taking a wide range of testing measures in response to the COVID-19 outbreak, according to the University of Oxford COVID-19 Government response tracker [45]. The data show that testing policy is not standardized and varies substantially across African countries: on 27 May 2020 (50%) countries were testing only those who have symptoms and meet specific criteria (i.e., key workers, persons admitted to hospital, encountered a known case or returned from overseas); 14 (26%) countries were testing anyone showing COVID-19 symptoms; nine (17%) countries were implementing Open public testing (e.g., "drive through" testing available to asymptomatic people); and four (7%) countries had no testing policy [46]. There is a high possibility that infections are far higher than reported [47]. Publicly available testing data suggests enormous differences in testing capacity and case identification across Africa. As of 29 June 2020, the positivity rate for COVID-19 testing in Africa varied from 0.4% in Uganda to 25.7% in Nigeria [48]. Some countries, like Australia, South Korea, and Uruguay have test positivity rates of less than 1%. This implies that it takes hundreds, or even thousands of tests to find one case in these countries. Countries such as Mexico and Nigeria [48], on the other hand, have positivity levels of 20–50%, or even more. According to a WHO report, countries with high positivity rates are unlikely to be testing widely enough to find all cases. WHO recommends a positive rate of around 3% to 12% as a general benchmark of adequate testing [46].

There are, however, a number of limitations to this analysis. Serosurveys indicate a strong association between age and IFR. A systematic review found an exponential relationship between age and IFR for COVID-19 [49]. The calculated age-specific IFR for children and younger adults was relatively low (e.g., 0.002% at age 10 and 0.01% at age 25), but it gradually increased to 0.4% at age 55, 1.4% at age 65, 4.6% at age 75, and 15% at age 85. Furthermore, the study showed that about 90% of the variation in population IFR

across geographical areas was attributed to variations in the population's age structure and the degree to which the virus was exposed to relatively susceptible age groups. According to another study, COVID-19 IFR increases exponentially with age, as well as pneumonia and influenza [50]. Across the adult age group, COVID-19 IFR levels were 2.8 to 8.2 times greater than pneumonia and influenza [51]. While age-disaggregated data for all age groups were not available at the time of our study, the authors estimated the overall cumulative number of infections used in the IFR measurement using a dichotomous age variable. We acknowledge that the dichotomous age variable used in this study would not have been sufficient to explain the difference in IFRs. We recommend epidemiological studies to determine age-specific infection fatality rates for COVID-19 in Africa. Public health response variables (i.e., timing of index case identification and data on travel restrictions and lockdowns), and other important risk-factors, particularly cancer, chronic kidney disease, obesity, and sickle cell disease were not available for analysis. The variability in testing and cause-of-death data across African countries might have impacted the results.

## 5. Conclusions

Assessing the infection fatality rate of COVID-19 is crucial to determine the appropriateness of mitigation strategies and to enable planning for healthcare needs as epidemics unfold. Without population-based serologic studies in Africa, it is not yet possible to know what proportion of the population has been infected with COVID-19. Our study shows that Bayesian modeling is a helpful tool that can account for missed cases, such as those untested due to a country's low testing capacity, and the mild cases that are potentially missed in current surveillance activities. Using an estimated number of cumulative infections, the IFR can be calculated. In many countries in Africa, owing to weaker health-care systems, informal settlements, overcrowded cities and public transportation, and a lack of clean water and sanitation, the current approaches to self-protection, social distancing, and containment as measures to control the outbreak may not be viable. Scaling up surveillance efforts and growing COVID-19 research and testing capability across Africa may help to provide a deeper understanding of how the pandemic is advancing, and to define hotspots for targeted and pooled testing, case isolation, and early treatment. Our estimates of the underlying infection fatality rate of this virus will inform assessments of health effects likely to be experienced in different countries, and thus decisions around appropriate mitigation policies and strategies that should be adopted.

**Author Contributions:** A.A.O. conceived of the study including design and method. He was the principal in data management, analyzed the data, and drafted the article. A.K., C.O., G.O. contributed to the validation of data and reviews of the first draft. J.E. and O.K. are joint last authors and performed critical reviews of the second and final versions of the manuscript. All authors have read and agreed to the published version of the manuscript.

**Funding:** This research received no specific grant from any funding agency in the public, commercial or not-for-profit sectors. O.K. was supported by a professorship grant from the Swiss National Science Foundation (grant no. 163878) and a Swiss National Science Foundation project grant (grant no. 320030_192452).

**Institutional Review Board Statement:** Not applicable.

**Informed Consent Statement:** Not applicable.

**Data Availability Statement:** This dataset and code for analyses are available on the following website: https://covid.ourworldindata.org/data/owid-covid-data.csv (accessed on 28 April 2021).

**Conflicts of Interest:** The authors declare no conflict of interest.

## Appendix A

*Appendix A.1. Sensitivity Analysis of the Infection Fatality Rates of the Posterior Summary Statistics, 30 May 2020*

**Table A1.** Posterior Summary Statistics.

| Posterior Summary Statistics | Mean | | | Maximum | | |
|---|---|---|---|---|---|---|
| | 75% Cred. Interval | 90% Cred. Interval | 95% Cred. Interval | 75% Cred. Interval | 90% Cred. Interval | 95% Cred. Interval |
| Cumulative COVID-19 Infections Estimated (as of 30 May 2020) | 47,366 [44,565, 50,167] | 56,839 [54,038, 59,640] | 63,154 [60,353, 65,955] | 1,265,159 [1,262,358, 1,267,960] | 1,518,191 [1,515,390, 1,520,992] | 1,686,879 [1,684,078, 1,689,680] |
| Calculated IFR (as of 30 May 2020) | 8.28% | 6.90% | 6.21% | 0.31% | 0.26% | 0.23% |
| Total Number of Infections Per confirmed Case (as of 30 May 2020) | 0.35 | 0.42 | 0.47 | 9.36 | 11.24 | 12.48 |

*Appendix A.2. Posterior Predictive Checks for Convergence Across All Model Parameters*

The mean and variance parameters for the trace plots mix very well. Autocorrelation is essentially negligible for all positive lags. The kernel density estimates based on the first and second halves of the sample are remarkably similar to each other and are close to the overall density estimates. The histogram and kernel density plots resemble the shape of an expected inverse-gamma distribution.

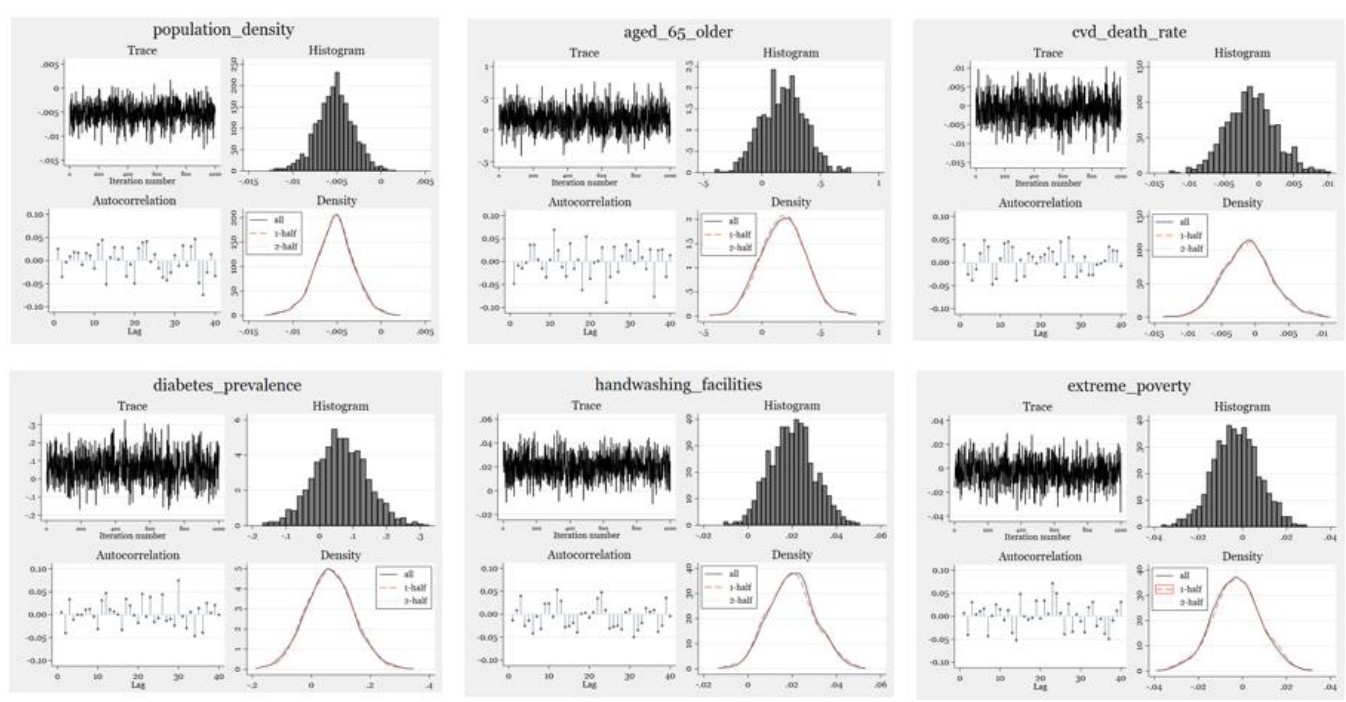

**Figure A1.** Model Evaluation.

*Appendix A.3. Posterior Predictive Summary for Test Statistics*

**Table A2.** Test Statistics.

| | Posterior Predictive Summary | | | MCMC Sample Size = 1000 | | |
|---|---|---|---|---|---|---|
| | **T** | **Mean** | **Std. Dev.** | **E(T_Obs)** | **P(T ≥ T_Obs)** | |
| **Mean** | 6.749488 | | 0.3085233 | 6.775585 | 0.462 | |
| **Min** | 2.818528 | | 1.005077 | 3.218876 | 0.389 | |
| **Max** | 10.89979 | | 1.062004 | 10.28329 | 0.694 | |

Note: $P(T \geq T\_obs)$ close to 0 or 1 indicates lack of fit.

*Appendix A.4. WHO COVID-19 Transmission Classification Type*

Imported/Sporadic cases: Cases detected in the past 14 days are all imported, sporadic (e.g., laboratory-acquired or zoonotic), or are all linked to imported/sporadic cases, and there are no clear signals of further locally acquired transmission. This implies a minimal risk of infection for the general population.

Clusters of cases: Cases detected in the past 14 days are predominantly limited to well-defined clusters that are not directly linked to imported cases, but are all linked by time, geographic location, and common exposures. It is assumed that there are a number of unidentified cases in the area. This implies a low risk of infection to others in the wider community if exposure to these clusters is avoided.

Community transmission: Encompasses a range of levels from low to very high incidence, as described below and informed by a series of indicators described in the aforementioned guidance. As these subcategorizations are not currently collated at the global level, but rather intended for use by national and sub-national public health authorities for local decision-making, community transmission was disaggregated in this information product.

CT1: Low incidence of locally acquired, widely dispersed cases detected in the past 14 days, with many of the cases not linked to specific clusters; transmission may be focused in certain population sub-groups. Low risk of infection for the general population.

CT2: Moderate incidence of locally acquired widely dispersed cases detected in the past 14 days; transmission less focused in certain population sub-groups. Moderate risk of infection for the general population.

CT3: High incidence of locally acquired, widely dispersed cases in the past 14 days; transmission widespread and not focused in population sub-groups. High risk of infection for the general population.

CT4: Very high incidence of locally acquired widely dispersed cases in the past 14 days. Very high risk of infection for the general population.

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
