# Peer review of "Estimates of the COVID-19 Infection Fatality Rate for 48 African Countries: A Model-Based Analysis"

_2673-8430, doi:10.3390/biomed1010005_

Round 1

Reviewer 1 Report

The study of Onovo et al. is aimed to assess the SARS-CoV-2 infection fatality rate of data from 48 African countries. The authors analyzed time series data of SARS-CoV-2 infection outbreak from 15th February 2020 to 30th May 2020 using a Bayesian prediction model based on the technique of Monte Carlo. Results showed a IFR in Africa was estimated to be 0.23% (95%CI: 0.14% to 0.33). Authors concluded that The infection fatality rate of COVID-19 can vary substantially across different locations, and this may reflect differences in demographics, underlying health issues in the population, capacity of the healthcare system, positive health seeking behavior, as well as other factors.

Article is original and well written. The statistical analysis is well assessed

I have some issue to report:

  • the time interval considered in the observation is short in relation to the duration of the pandemic. Please clarify
  • Authors reported in text “We used publicly documented COVID-19 datasets created by Our World in Data (https://github.com/owid/covid-19-data/tree/master/public/data)”. However, the link indicated refers to a data sharing system and the precise source of these is difficult to find. Please specify the official source of the data used.
  • Please add a reference for the Bayesian statistical model use in the present study

Author Response

Point 1: the time interval considered in the observation is short in relation to the duration of the pandemic. Please clarify

Response 1: This study's analysis was completed on the 30th of May 2020. The authors wanted to figure out the real cumulative COVID-19 infection rate (denominator for calculating infection fatality rate) based on verified coronavirus infections and deaths recorded between February 15, 2020, and May 30, 2020, when the study was conducted.

Point 2: Authors reported in text “We used publicly documented COVID-19 datasets created by Our World in Data (https://github.com/owid/covid-19-data/tree/master/public/data)”. However, the link indicated refers to a data sharing system and the precise source of these is difficult to find. Please specify the official source of the data used.

 Response 2: On line 83, the specific URL and/or data source for the COVID-19 dataset utilized in this work is included. https://covid.ourworldindata.org/data/owid-covid-data.csv”.

Point 3: Please add a reference for the Bayesian statistical model use in the present study

Response 3: The reference for the Bayesian model utilized in this study has been included on line 189. The STATA website has a link to the Bayesian regression model used in this study at https://www.stata.com/features/overview/bayesian-predictions/.

Reviewer 2 Report

The paper titled “Estimates of the COVID-19 Infection Fatality Rate for 48 African Countries: a model-based analysis” Examine the global data from 48 African countries to estimate the SARS-CoV-2 infection fatality rate. Authors use a Bayesian parametric model making different Assumptions that are clearly exposed in the paper. After the analysis they conclude that the infection fatality rate of COVID-19 can vary substantially across the 32 different locations studied, and this may reflect differences in demographics, underlying health issues in the population, capacity of the healthcare system, positive health seeking behavior, as well as other factors.

Only minor points must be clarifying by the authors

  • In paragraph between lines 70-83 Author says that representative seroprevalence studies provide an important opportunity to estimate the number of infections in a community, and when combined with death counts can lead to robust estimates of the IFR. Some studies made by other authors are listed in this paragraph, but they do not say the model applied by these authors. Are there other similar studies make in other disease to validate this study? Or some similar study was made previously using the Bayesian parametric model?
  • Throughout the text there are paragraphs with different font, please correct them.

Author Response

Point 1: In paragraph between lines 70-83 Author says that representative seroprevalence studies provide an important opportunity to estimate the number of infections in a community, and when combined with death counts can lead to robust estimates of the IFR. Some studies made by other authors are listed in this paragraph, but they do not say the model applied by these authors. Are there other similar studies make in other disease to validate this study? Or some similar study was made previously using the Bayesian parametric model?

Response 1: Thank you for pointing this out. The authors have added additional language to lines 67-69 to give information on the analytical model utilized by another research. Recent serosurvey in the canton of Geneva, Switzerland using a Bayesian regression model estimated a population-wide IFR of 0.64% (0.38–0.98) [13]. Gudbjartsson et al., recently published a research that estimated the prevalence of COVID-19 fatalities in Iceland using Bayesian analysis” [14].

Point 2: Throughout the text there are paragraphs with different font, please correct them.

Response 2: The manuscript's paragraphs with various fonts have been edited and replaced with Times New Roman font in size 9.

Round 2

Reviewer 1 Report

The changes med by the authors are relevant and sufficient to allow the paper publication